# Genome-Wide Identification MIKC-Type MADS-Box Gene Family and Their Roles during Development of Floral Buds in Wheel Wingnut (*Cyclocarya paliurus*)

**DOI:** 10.3390/ijms221810128

**Published:** 2021-09-19

**Authors:** Yinquan Qu, Weilong Kong, Qian Wang, Xiangxiang Fu

**Affiliations:** 1Co-Innovation Center for Sustainable Forestry in Southern China, Nanjing Forestry University, Nanjing 210037, China; qyquan@njfu.edu.cn (Y.Q.); 15764380756@163.com (Q.W.); 2Shenzhen Branch, Guangdong Laboratory of Lingnan Modern Agriculture, Genome Analysis Laboratory of the Ministry of Agriculture and Rural Affairs, Agricultural Genomics Institute at Shenzhen, Chinese Academy of Agricultural Sciences, Shenzhen 518120, China; Asuraprince@126.com

**Keywords:** *Cyclocarya paliurus* (Batal.) Iljinskaja, MIKC-type MADS-box gene family, expression pattern, floral development

## Abstract

MADS-box transcription factors (TFs) have fundamental roles in regulating floral organ formation and flowering time in flowering plants. In order to understand the function of MIKC-type MADS-box family genes in *Cyclocarya paliurus* (Batal.) Iljinskaja, we first implemented a genome-wide analysis of MIKC-type MADS-box genes in *C. paliurus*. Here, the phylogenetic relationships, chromosome location, conserved motif, gene structure, promoter region, and gene expression profile were analyzed. The results showed that 45 MIKC-type MADS-box were divided into 14 subfamilies: BS (3), AGL12 (1), AP3-PI (3), MIKC* (3), AGL15 (3), SVP (5), AGL17 (2), AG (3), TM8 (1), AGL6 (2), SEP (5), AP1-FUL (6), SOC1 (7), and FLC (1). The 43 MIKC-type MADS-box genes were distributed unevenly in 14 chromosomes, but two members were mapped on unanchored scaffolds. Gene structures were varied in the same gene family or subfamily, but conserved motifs shared similar distributions and sequences. The element analysis in promoters’ regions revealed that MIKC-type MADS-box family genes were associated with light, phytohormone, and temperature responsiveness, which may play important roles in floral development and differentiation. The expression profile showed that most MIKC-type MADS-box genes were differentially expressed in six tissues (specifically expressed in floral buds), and the expression patterns were also visibly varied in the same subfamily. *CpaF1st24796* and *CpaF1st23405*, belonging to AP3-PI and SEP subfamilies, exhibited the high expression levels in PA-M and PG-F, respectively, indicating their functions in presenting heterodichogamy. We further verified the MIKC-type MADS-box gene expression levels on the basis of transcriptome and qRT-PCR analysis. This study would provide a theoretical basis for classification, cloning, and regulation of flowering mechanism of MIKC-type MADS-box genes in *C. paliurus*.

## 1. Introduction

MADS-box gene family, one of the most extensively studied transcription factors (TFs), have fundamental roles in developmental control and signal transduction in plants, animals, and fungi [1]. According to the sequence structure, MADS-box gens have been divided into two super clades: type I and type II. Among them, type I encodes one SRF-like MADS domain with a simple structure, containing only 1 to 2 exons, whereas type II constitutes MIKC^C^ and MIKC^∗^ in plants encodes a MEF2-like and MIKC type MADS domain with semi-conservative K-box (keratin-like domain) and poorly conservative I-box (intervening domain) and variable C-terminal domain (C-terminal region) [2,3]. MIKC-type MADS-box genes are best known for their functions: regulating the development of floral organs and fruits, and controlling flowering time and gametophytic cell division, photosynthesis, and nutrient metabolism [4]. The MIKC^C^-type genes have been subdivided into 13 groups, namely, AG (class C genes), AGL6, AGL12, AP3-PI (class B genes), Bs, SOC1, SVP, SEP (class E genes), AGL17, AP1-FUL (class A genes), AGL15, FLC, and TM8 [5]. Among of them, A and E genes regulate sepal development; A, B, and E genes jointly regulate petal development; B, C, and E genes jointly regulate stamen development; C and E genes regulate carpel development; and D and E genes regulate ovules development [6].

*Cyclocarya paliurus* (Batal.) Iljinskaja, a unique species growing in southern China, is the sole species in the genus *Cyclocarya* Iljinskaja (Juglangdaceae) [7]. It is mainly distributed at an altitude of 420–2500 m in the subtropical mountainous areas of Jiangxi, Zhejiang, Anhui, Fujian, Hubei, Sichuan, Guizhou, and Guangxi provinces of China, among other areas. [8]. As a Chinese native medicinal plant, its leaves can eradicate fever and detoxify and relieve pain, and thus it was widely used for making medicinal tea in ancient China [9]. Modern pharmacological studies have found that many potential bioactive compounds (triterpenoids, polysaccharides, flavonoids, etc.) in the leaves of *C. paliurus* have beneficial effects on antihypertensive activity, hypoglycemic activity, hypolipidemic activity, and antioxidant activity [10]. Due to the lack of breakthroughs in its asexual reproduction technology, sexual reproduction is still the main means of reproduction. Nevertheless, heterodichogamy with asynchronous flowering in *C. paliurus* [7] possesses two temporally complementary morphs, protandry (PA) or protogyny (PG), resulting in the low seed setting rate and high seed empty rate in its natural state, so as to limit resource expansion. To understand above area better, especially on molecular mechanism of *C. paliurus* flower bud differentiation, further studies on the MIKC-type MADS-box gene family at the molecular level are encouraged. Therefore, this study primarily aimed to identify and characterize MADS-box genes in *C. paliurus* and explore the roles of these genes during floral development stages.

In this study, we provide the first overview of the MADS-box gene family in *C. paliurus*, which will be used to lay the foundation for future functionality studies, particularly the MADS-box proteins that likely play important roles in heterodichogamy of *C. paliurus*. We also quantified transcript levels of MIKC-type MADS-box genes at floral bud germination using transcriptome and quantitative real-time PCR (qRT-PCR) analysis.

## 2. Results

### 2.1. Identification of the MIKC-Type MADS-Box Gene Family in C. paliurus

MIKC-type MADS-box genes were identified from the *C. paliurus* genome using the HMM program for the K-box domain search and the local BLASTP search with *Arabidopsis* MADS-box gene sequences. As a result, 45 identified MIKC-type MADS-box genes were serially named as *CpMADS01* through *CpMADS45* for convenience (Table 1). In addition, the results showed that the MADS-box genes varied substantially in the length of mRNA transcripts and their encoded protein sequences. The length of 45 *CpMADS* mRNA products (CDS) ranged from 474 to 1350 bp and that of translated protein sequences varied from 157 to 449 amino acids. The amino acid sequence of *CpMADS25* was the shortest, with only 157 amino acid residues; *CpMADS18* was the longest, with 449 amino acid residues. The theoretical isoelectric point (PI) was between 4.99 (*CpMADS05*) and 9.96 (*CpMADS22*). The molecular mass of MADS-box members ranged from 18.12 (*CpMADS25*) to 50.77 kDa (*CpMADS18*) (Table 1).

### 2.2. Chromosome Location and Synteny Analysis of MIKC-Type MADS-Box Genes

All MIKC-type MADS-box genes except for *CpMADS44* and *CpMADS45* in *C. paliurus* were mapped to 14 chromosomes (Chrs) (Figure 1). The maximum number of MADS-box genes per chromosome was found on Chr02 and Chr13, both with seven MADS-box genes. Six genes were located on Chr01; four genes were found on Chr06; and three genes each were located on Chr04, Chr05, and Chr07. Chr10, Chr11, Chr12, and Chr14 encompassed the minimum numbers of MADS-box genes, with only one each. Among MIKC-type MADS-box genes from *C. paliurus*, a total of 127 pairs of segmental duplicated genes were identified, as shown in Appendix A, and five pairs of tandem duplicated genes (*CpaF1st02480* and *CpaF1st02481*, *CpaF1st23405* and *CpaF1st23406*, *CpaF1st34609* and *CpaF1st34610*, *CpaF1st34822* and *CpaF1st34828*, *CpaF1st46308* and *CpaF1st46310*) were observed, as shown in Figure 1.

Syntenic analysis was performed for three plants (*Arabidopsis thaliana*, *Populus trichocarpa*, and *Juglans regia*) to investigate the evolutionary relationship of MIKC-type MADS-box genes among these three species (Figure 2). A total of 23, 7, and 6 orthologous gene pairs of orthologous MADS-box genes were identified between *C. paliurus* and *J. regia*, *C. paliurus* and *P. trichocarpa*, and *C. paliurus* and *A. thaliana*, respectively (Figure 2). In contrast, 13 collinear gene pairs identified between *C. paliurus* and *J. regia* were found between *C. paliurus* and *P. trichocarpa*/*A. thaliana*, 6 collinear gene pairs identified between *C. paliurus* and *P. trichocarpaa* were presented between *C. paliurus* and *J. regia*/*A. thaliana*, and all 6 collinear gene pairs identified between *C. paliurus* and *A. thaliana* were detected between *C. paliurus* and *P. trichocarpaa*/*J. regia* (Figure 2). Moreover, we also noticed that there were five common collinear gene pairs in each of the following species: *C. paliurus*, *J. regia*, and *A. thaliana* (Figure 2).

### 2.3. Phylogeny, Structural, and Conserved Motifs Analysis of MIKC-Type MADS-Box Genes

To identify the evolutionary relationship of MIKC-type MADS-box proteins, we constructed a neighbor-joining (NJ) phylogenetic tree on the basis of the full-length protein sequences (Figure 3). The MADS-box proteins were divided into 14 subfamilies in *C. paliurus* (Figure 3). Among the 45 MIKC-type MADS-box members, only three members, namely, *CpaF1st12920*, *CpaF1st25075*, and *CpaF1st14025*, were classified as MIKC*. The other 42 genes were classified as MIKC^C^, which were divided into 13 subfamilies. The subfamily SOC1 with seven MADS-box family members were the largest clade. In contrast, the subfamilies AGL12, TM8, and FLC contained the smallest number of MADS-box families with one member. Moreover, the numbers of the subfamilies BS, AP3-PI, AGL15, SVP, AGL17, AG, AGL6, SEP, and AP1-FUL were 3, 3, 3, 5, 2, 3, 2, 5, and 7, respectively.

The gene structure of 45 MIKC-type MADS-box genes in *C. paliurus* was analyzed (Figure 4A,B). We observed that the gene structure of MIKC-type MADS-box members was relatively variable as the exon number ranged from 4 to 26. Among them, *CpaF1st12920* of MIKC* subfamily contained the maximum exons of up to 26, which was obviously different from the structures of the other two subfamily members (the exon numbers were 8 and 19, respectively). Similarly, the *CpaF1st23406* of AP1-FUL subfamily contained 15 exons, but members in other subfamily contained only seven to eight exons. By contrast, the exon numbers of BS subfamily members were extremely few with four to seven exons in each gene. These results indicated that the MIKC-type MADS-box genes structures were also visibly varied, even in the same gene family or subfamily.

A total of 10 different motifs were detected in MIKC-type MADS-box gene family of *C. paliurus* (Figure 4C). A total of 32 of 45 MADS-box genes contained at least 4 main motifs (motifs 1, 2, 4, and 5). Motif 9 was specific to the subfamily AGL15 proteins CpaF1st16532 and CpaF1st11746. Motif 8 only existed in AP1-FUL. Motif 10 was shared by subfamilies FLC, SOC1, AGL15, and SVP. Each member belonging to SEP and AGL6 subfamilies shared the same seven motifs among the gene structures. The C-terminal of *C. paliurus* MADS-box family members consisted of five motifs: motif 5, motif 6, motif 7, motif 8, and motif 5. Moreover, the N-terminal was composed of only two motifs: motif 1 and motif 5 (Figure 4C). This evidence collectively supported that the same subfamily consists of almost identical conserved motifs.

### 2.4. Estimation of Nonsynonymous (Ka) and Synonymous (Ks) Substitution Rates and Ka/Ks Values

To explore the selection pressure experienced by MIKC-type MADS-box genes after the duplication during evolution, we calculated the substitution ratios of nonsynonymous (Ka) versus synonymous substitution (Ks) for each paralogous pair (Table 2). We identified a total of five pairs of tandem replications in *C. paliurus*. The Ka/Ks ratio ranged from 0.2165 (CpaF1st02480 and CpaF1st02481) to 1.3276 (CpaF1st46308 and CpaF1st46310) with an average of 0.6473. The ratios of Ka/Ks of three pairs of paralogous MADS-box gene pairs (CpaF1st02480 and CpaF1st02481, CpaF1st23405 and CpaF1st23406, and CpaF1st46308 and CpaF1st46310) were less than 1, suggesting that these genes are under purifying selection. The value of Ka/Ks (CpaF1st34609 and CpaF1st34610, and CpaF1st34822 and CpaF1st34828) were greater than 1, suggesting positive selection for both duplicates in the two gene pairs. The results indicated that most of the gene pairs have evolved under purifying selection pressure and that the duplication might play an important role in the expansion of MADS-box gene family in *C. paliurus*.

### 2.5. Promoter Region Analysis of MIKC-Type MADS-Box Genes

To identify cis-acting elements involving *C. paliurus* flower development, we analyzed the promoter sequences of MIKC-type MADS-box genes. In addition to the cis-acting element, which commonly exists in eukaryotes including TATA-box and CAAT-box, the most predominant cis-acting elements in the promoters’ regions of MADS-box family genes were associated with light, phytohormone, and temperature responsiveness (Figure 5). Additionally, the common cis-acting elements, such as ARE, G-box, ABRE, CGTCA-motif, and TGACG-motif, were frequently identified in the majority of MADS-box members, with at least 30 genes containing the above elements. By contrast, we also found several cis-elements (Sp1, P-box, A-box, GARE-motif, and MRE), which were distributed in minority members, with at least 16 genes containing the above elements. Moreover, a number of cis-elements were related to light responsiveness through G-box, G-Box, GT1-motif, MBS, Sp1, and MRE. The CGTCA-motif and TGACG-motif were associated with methyl jasmonate (MeJA) responsiveness. The LTR was involved in low-temperature responsiveness. The ABRE and GARE-motif were related to abscisic acid and gibberellin responsiveness, respectively. The CAT-box was related to regulation of meristem expression (Appendix A). These results indicated that different components of cis-elements in MADS-box genes might effectively perform different biological functions during flower development and differentiation.

### 2.6. Expression Profiles of MIKC-Type MADS-Box Genes among Different Tissues

To investigate potential functions of MIKC-type MADS-box genes among six tissues of *C. paliurus* during the floral initiation, we performed RNA-seq analysis to examine expression patterns of 45 MADS-box genes (Appendix A). Except for SOC1 and SVP subfamilies, which were highly expressed not only in floral buds but also in leaves, most of the other subfamilies were expressed specifically in floral buds. Further, we found that the expression levels of AP1-FUL, AP3-PI, AG, and SEP subfamilies belonging to A, B, C, and E modules, respectively, were higher than the other 10 subfamily members. However, the AP1-FUL (A module) subfamily *CpaF1st12808* gene and AG (C module) subfamily *CpaF1st24126* were weakly expressed in each floral bud. These results indicated that the MIKC-type MADS-box gene expression levels were also visibly varied in the same gene family, and even in the same subfamily. Interestingly, the expression level of AP3-PI (B) subfamily *CpaF1st24796* in PA male floral buds (PA-M) was higher than in PA female floral buds (PA-F) and PG male floral buds (PG-M). These results strongly suggest that the high expression level of *CpaF1st24796* could promote the developmental priority of the male flower, and eventually present as a PA mating type. On the other hand, the expression of SEP subfamily *CpaF1st23405* in PG female floral buds (PG-F) was higher than in PA female floral buds (PA-F) and PG male floral buds (PG-M), which may preferentially promote female floral bud development and eventually contribute to its PG mating type. Moreover, most gene pairs of tandem replications also showed the different expression patterns. For instance, *CpaF1st02481* was strongly expressed in four floral buds, but *CpaF1st02480* was weakly expressed in them (Appendix A). In addition, a total of nine ABCE module MADS-box genes, namely, SEP subfamily *CpaF1st02480*, *CpaF1st02666*, and *CpaF1st12676* genes; AG subfamily *CpaF1st08032* and *CpaF1st38580* genes; AP3-PI subfamily *CpaF1st24796* gene; and AP1-FUL subfamily *CpaF1st29740* and *CpaF1st18733* genes, were selected for qRT-PCR analysis. Generally, the relative expressions of selected genes were consistent with FPKM values from RNA-seq, indicating the expression profiles of our transcriptomic results were reliable (Figure 6).

## 3. Discussion

MIKC-type MADS-box transcription factors are one of the notable protein families in flowering plants involved in flower induction, floral initiation, and floral morphogenesis [11]. In this study, a genome-wide identification of the MIKC-type MADS-box protein family was carried out in *C. paliurus*. The identified 45 *C. paliurus* MIKC-type MADS-box proteins were classified into 14 distinct subfamilies, which is also consistent with the classification and identified results of *A. thaliana* [12]. Comparing the number of each subfamily, we found that only one and three MIKC-type MADS-box members in subfamilies FLC and MIKC* were far less than six in *A. thaliana,* respectively (Figure 3). On the contrary, six and five members in subfamily AP1-FUL and SVP were more than three and two in *A. thaliana,* respectively (Figure 3). Previously, the identification of MIKC-type MADS-box genes have been reported in many other plants, such as 38 members in *Vitis vinifera* [6], 39 members in *Prunus persica* [13], 36 members in *Punica granatum* [14], 201 members in *Triticum aestivum* [15], and 110 members in *Gossypium hirsutum* [16]. The differences in the number of MIKC-type MADS-box members between *C. paliurus* and *T. aestivum* or *G. hirsutum* may be due to the fact that *T. aestivum* and *G. hirsutum* experienced gene duplication events, including genome-wide duplication, tandem duplication, and fragment duplication. Comparing the number of duplicated genes of MIKC-type MADS-box genes between *C. paliurus* and *A. thaliana*, we found the 132 (127 segmental duplicated and 5 tandem duplicated genes) gene pairs to be more than 81 in *A. thaliana* [12], which may promote the expansion of the MIKC-type MADS-box gene family. Moreover, the 583.45 Mb assembled genome size of *C. paliurus* was normally larger than 119.67 Mb in *A. thaliana* [17]. Comparison of the MIKC-type MADS-box gene number between *C. paliurus* and *A. thaliana* showed that there was no significant correlation between genome size and number of members.

The structures and conserved motifs of each subfamily genes were analyzed on the basis of our classification results from phylogenetic tree analysis. The MIKC-type MADS-box genes structures were visibly varied, even in the same gene subfamily, which is similar to the results reported in *Beta vulgaris* [18], but different from the results in *Punica granatum* [14] and citrus [19]. Motif analysis demonstrated that most closely related MADS-box proteins from the same subfamilies shared similar motif distributions and sequences. Similar results were also reported in *Gossypium hirsutum* [16] and *Lactuca sativa* [20]. On the basis of the expression analysis, most of the same subfamily genes showed the same expression patterns in different tissues, but small amounts of genes from the same subfamilies were also differentially expressed. For instance, AG subfamily genes *CpaF1st38580* and *CpaF1st08032* showed high expression levels in four types of floral buds, but *CpaF1st24126* presented contrary expression patterns. The study indicated that the same subfamily genes had divergent and differentiated expressions in terms of temporal and spatial characteristics during the process of evolution. These results may have been caused by the inherent need for paragenetic genes of the same origin to avoid functional redundancy and to produce subfunctions and new functions. In addition, most genes of paralogous MIKC-type MADS-box gene pairs in *C. paliurus* showed different expression patterns, indicating that *C. paliurus* may undergo multiple selection directions during the evolutionary process [21].

Promoter region analysis discovered that most of the components of cis-elements in MIKC-type MADS-box genes were associated with light, phytohormone, and temperature responsiveness. In addition, most MADS-box genes showed higher expression levels in floral buds than leaves. These results indicated that MIKC-type MADS-box genes might play an important role in flower development and differentiation, which was also supported by the results in *A. thaliana* [12]. As a typical heterodichogamous species, *C. paliurus* possesses two complementary morphs with asynchronous flowering (PG and PA mating types), which may effectively prevent from selfing, reduce intramorph inbreeding, and heavily contribute to the pattern of genetic diversity in the process of species evolution. Important as they are, the molecular mechanisms involving heterodichogamy remain unknown. In our study, the expression level of AP3-PI subfamily *CpaF1st24796* in PA-M was higher than in PA-F and PG-M. On the other hand, the expression of SEP subfamily *CpaF1st23405* in PG-F was higher than in PA-F and PG-M (Appendix A). These results strongly suggest that the high expression of *CpaF1st24796* or *CpaF1st23405* in PA-M or PG-F may promote male or female flower development priority, eventually presenting as the PA or PG mating type. These two gene functions were also consistent with previous reports on *A. thaliana* [22], *Rosa chinensis* [23], and *Populus trichocarpa* [24].

## 4. Materials and Methods

### 4.1. Plant Materials, cDNA Synthesis, and Transcriptome Sequencing

The female flowers, male flowers, and leaves in various protandry (PA) and protogyny (PG) individuals were sampled at germination stage (24 March 2019) from germplasm bank of *C. paliurus*, which is located in Baima experimental base of Nanjing Forestry University (31°35′ N, 119°09′ E), Nanjing, Jiangsu Province, China. Collected samples were immediately frozen in liquid nitrogen and stored at −80 °C. Each tissue sample contained three biological replicates, and each replica was composed of tissues from 3 individuals. Total RNA was extracted from various tissues using E.Z.N.A Plant RNA Isolation Kit (OMEGA, USA). RNA concentration and purity were determined by NanoDrop spectrophotometer 2000C (Thermo Fisher Scientific, Waltham, MA, USA), and RNA integrity was confirmed by 1.0% agarose gel electrophoresis. The RNAs with the values of OD_260_/OD_280_ within 1.8–2.0 were reverse transcribed to cDNA using PrimeScript™ RT reagent kit with gDNA Eraser (TaKaRa, China), according to the instructions of the manufacturer. Then, the cDNAs were diluted at 1:10 with RNase-free water and stored at −20 °C for qRT-PCR analyses. All cDNA libraries were loaded onto an Illumina HiSeqTM2000 system (2 × 100 bp read length), and the quality of raw reads was checked by FastQC [25]. The poor-quality sequences (the bases of Q-score < 10%), short sequences (<50 bp), primer sequences, and adapter sequences were removed and excluded from the raw reads.

### 4.2. Identification of MIKC-Type MADS-Box Genes in C. paliurus

The *C. paliurus* whole protein sequence was downloaded from the National Center for Biotechnology Information (NCBI) (http://www.ncbi.nlm.nih.gov/, accessed on 10 August 2021). To identify the MIKC-type MADS-box proteins from *C. paliurus*, we used *Arabidopsis* MADS-box protein sequences as a query [12] (Appendix A), and the local alignment search tool (BLASTP) [26] with E value less than 1e-10 was applied for global search of MIKC-type MADS-box proteins. After this initial screening, the HMM (hidden Markov model) profile for the Pfam MADS-domain (PF01486) was obtained from the pfam database (http://pfam.xfam.org/, accessed on 10 August 2021). Then, the HMM profile was used to search against *C. paliurus* proteomes using the HMMER v3.3.2 [27] with default parameters. The obtained sequences were filtered by the following two criteria: (1) incomplete sequences were removed if the sequence lengths were less than the length of conserved motif (33 bp); (2) sequences with ‘N’ >10 bp were also excluded from further analysis.

### 4.3. Chromosome Location and Synteny Analysis of MIKC-Type MADS-Box Genes

MIKC-type MADS-box genes of *C. paliurus* were exactly located on the chromosome regions using the MapGene2Chrom wb v2 tool (http://mg2c.iask.in/mg2c_v2.0/, accessed on 11 August 2021) on the basis of genome annotation downloaded from the NCBI database. Multiple Collinearity Scan toolkit (MCScanX) [28] was adopted to identify the synteny relationship of homologous MIKC-type MADS-box genes obtained from *C. paliurus* and other three species (*A. thaliana*, *P. trichocarpa*, and *J. regia*).

### 4.4. Prediction of Cis-Regulatory Elements of Promoter Region

The 2000 bp upstream sequences of *C. paliurus* MIKC-type MADS-box genes were extracted from whole-genome sequence and were analyzed using PlantCARE promoter analysis tool (http://bioinformatics.psb.ugent.be/webtools/plantcare/html/, accessed on 13 August 2021) with default parameters for the prediction of various cis-acting regulatory elements [29,30].

### 4.5. Estimation of Ka/Ks Values

The duplicated gene pairs belonging to tandem and segmental duplications were extracted from whole genome genes. Then, we calculated the Ka and Ks substitution rates by using KaKs_Calculator toolkit [31] on the basis of verified duplicated gene pairs of the MIKC-type MADS-box family.

### 4.6. Expression Analysis of Selected MIKC-Type MADS-Box Genes

After the reads’ quality control and adapter trimming, the clean RNA-seq reads were aligned to reference genomes using HiSAT2 with default parameters, and the expected number of FPKM fragments mapped were calculated using StringTie program [32] with default parameters.

Nine genes related to floral bud differentiation and development, as well as hormone-mediated signaling pathway, were selected for qRT-PCR analysis, including *CpaF1st02666*, *CpaF1st02480*, *CpaF1st12676*, *CpaF1st08032*, *CpaF1st38580*, *CpaF1st24796*, *CpaF1st29740*, and *CpaF1st18733*. Moreover, internal control 18S ribosomal RNA were selected according to Chen’s report [7]. The reaction conditions of PCR and methods of relative gene expression calculation were presented by Chen [7].

## Figures and Tables

**Figure 1 ijms-22-10128-f001:**
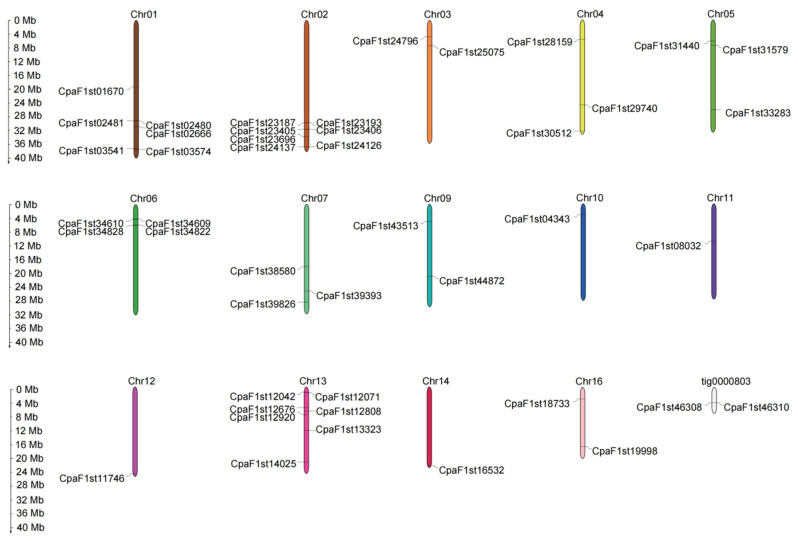
Chromosome distribution of MIKC-type MADS-box genes. *C. paliurus* chromosome number is indicated at the base of each chromosome with different colors. The MADS-box genes are marked at the approximate position on the chromosomes.

**Figure 2 ijms-22-10128-f002:**
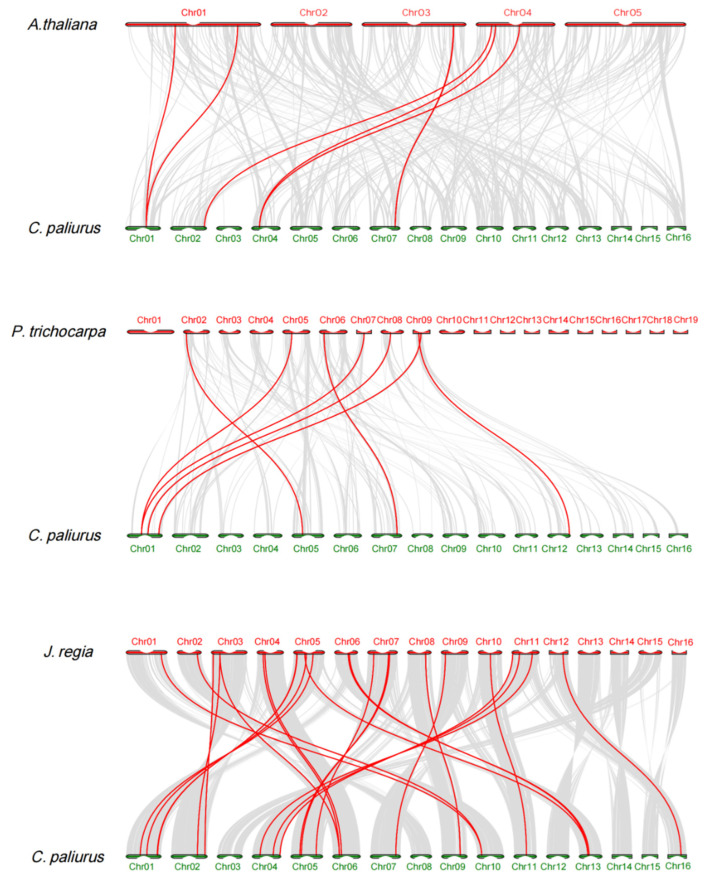
Synteny analysis of MIKC-type MADS-box genes between *C. paliurus* and three plant species. The collinear blocks between *C. paliurus* and other three plant genomes are shown as the gray lines in the background, while the syntenic MIKC-type MADS-box gene pairs are highlighted with red lines.

**Figure 3 ijms-22-10128-f003:**
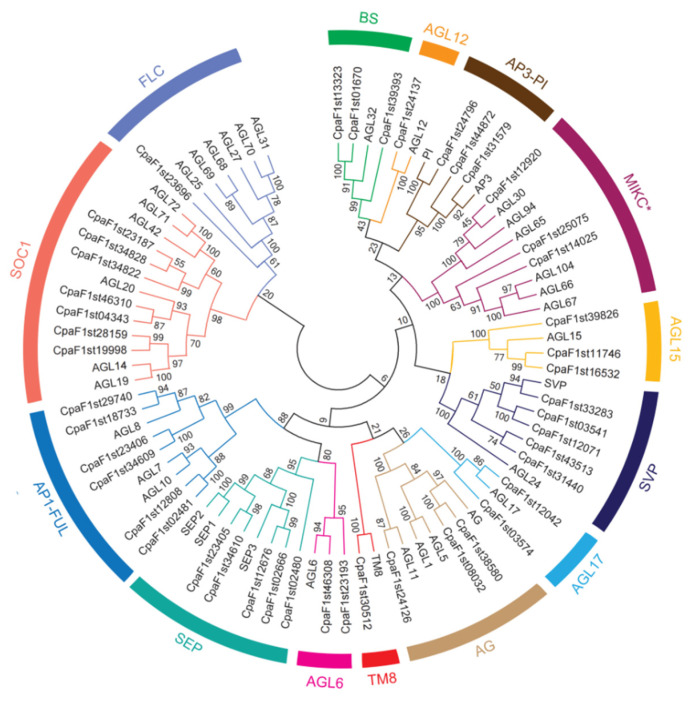
Phylogenetic analysis of MIKC-type MADS-box proteins in *C. paliurus* (45). A neighbor-joining (NJ) tree was constructed using 45 MADS-box sequences. The tree further clustered into 14 subfamilies, which are shown in different colors.

**Figure 4 ijms-22-10128-f004:**
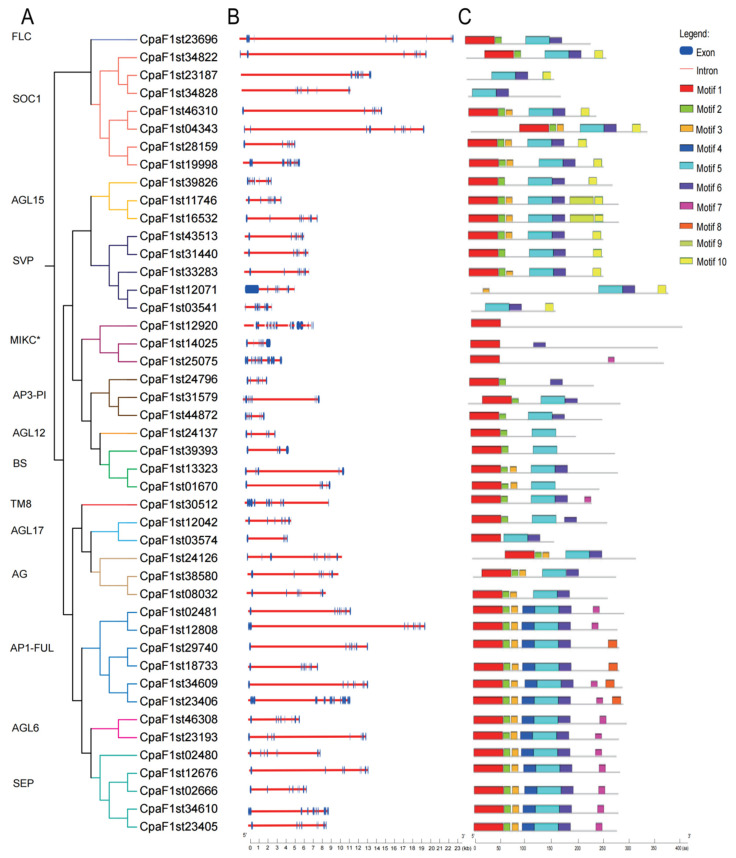
The phylogenetic relationship, gene structure, and motif compositions of MIKC-type MADS-box proteins. (**A**) The phylogenetic relationships of MADS-box proteins based on the NJ method. The various colors characterize the 14 subfamilies. (**B**) Gene structures of the MADS-box gene family was generated on the basis of the gene Structure Display Server. Exons (blue rectangles) and introns (pink lines) are marked along with their sequence length. (**C**) Motif in MIKC-type MADS-box proteins were predicted by MEME. The conserved motifs are represented by different colored boxes.

**Figure 5 ijms-22-10128-f005:**
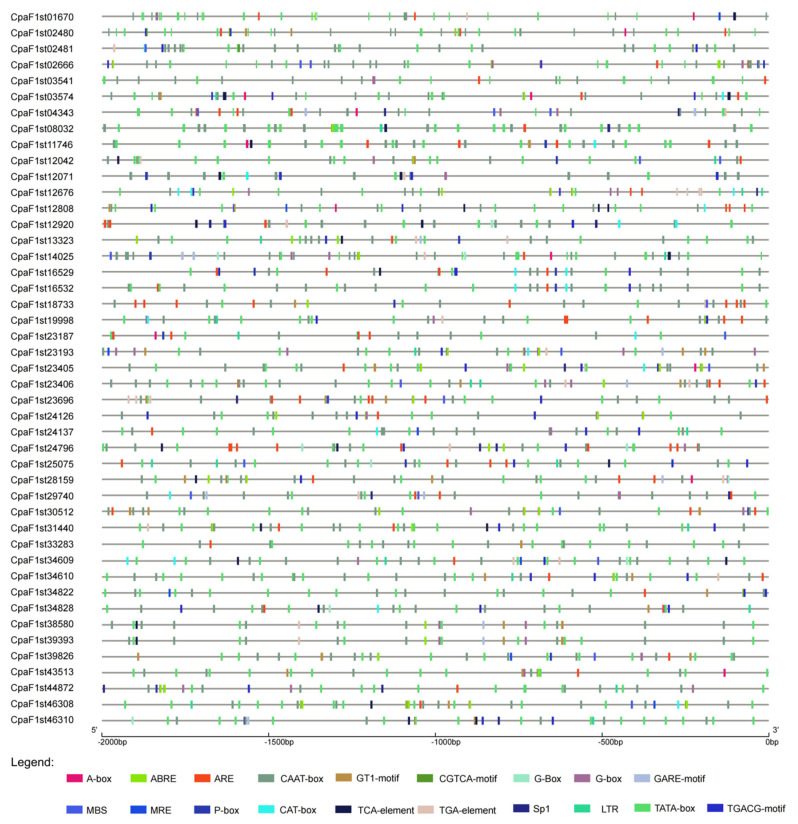
Cis-acting components of MIKC-type MADS-box genes in C. paliurus. All promoter sequences (2000 bp) were analyzed. The MADS-box genes are shown on the left. Scale bar at the base indicates length of promoter sequence. The functions of cis-acting element can be found in Appendix A.

**Figure 6 ijms-22-10128-f006:**
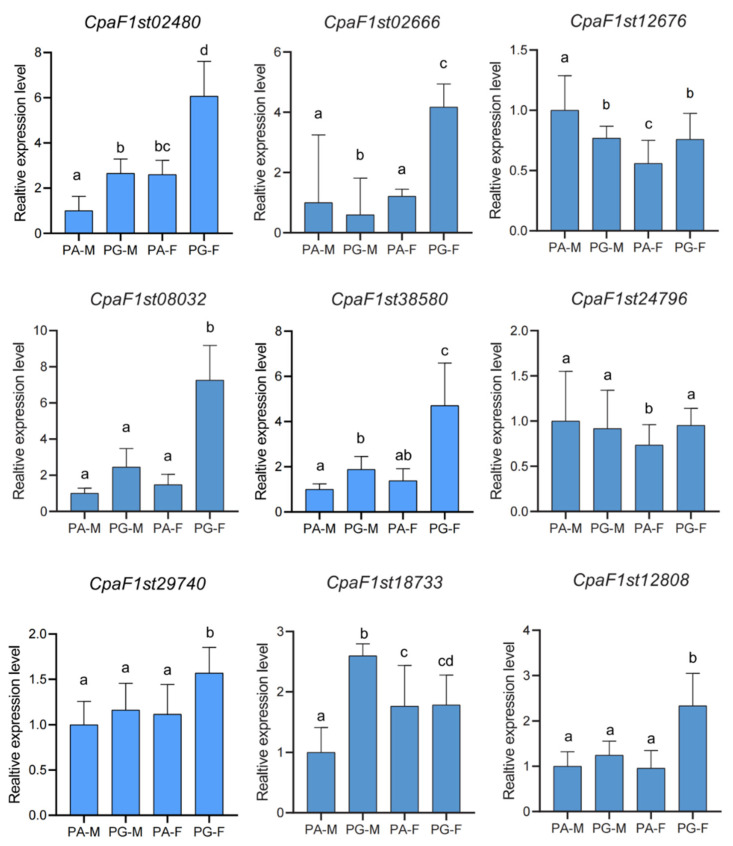
Comparisons of expression levels of nine MIKC-type MADS-box genes obtained by qRT-PCR analysis in different floral buds during germination in *C. paliurus*. PA-M: male floral buds in protandrous type; PG-M: male floral buds in protogynous type; PA-F: female floral buds in protandrous type; PG-F: female floral buds in protogynous type. All RT-qPCRs for unigenes were accomplished in triplicate, with two repeats per experiment. Error bars indicate SD, and different lowercase letters (a–d) represent significant differences among the three samples at *p* < 0.05.

**Table 1 ijms-22-10128-t001:** Physical and chemical properties of MIKC-type MADS-box genes in *C. paliurus* (CDS: coding DNA sequence; PI: isoelectric point; MW: molecular weight).

Gene Name	Gene ID	Exon Count	Chromosome Localization	Genomic Sequence (bp)	CDS (bp)	Amino Acid (aa)	PI	MW (kDa)
*CpMADS01*	*CpaF1st01670*	5	Chr1:22512748-22522152	9405	654	217	6.00	25.26
*CpMADS02*	*CpaF1st02480*	8	Chr1:34029459-34037773	8315	729	242	8.48	28.10
*CpMADS03*	*CpaF1st02481*	8	Chr1:34039378-34050977	11,600	771	256	9.23	29.73
*CpMADS04*	*CpaF1st02666*	7	Chr1:36013106-36019785	6680	738	245	8.88	28.09
*CpMADS05*	*CpaF1st03541*	7	Chr1:43429845-43432936	3092	483	160	4.99	19.28
*CpMADS06*	*CpaF1st03574*	4	Chr1:43721431-43725962	4532	549	182	9.37	20.72
*CpMADS07*	*CpaF1st23187*	16	Chr2:34639558-34653544	13,987	630	209	6.85	24.00
*CpMADS08*	*CpaF1st23193*	8	Chr2:34702417-34715817	13,401	744	247	9.07	28.66
*CpMADS09*	*CpaF1st23405*	16	Chr2:36737147-36746074	8928	735	244	8.73	27.97
*CpMADS10*	*CpaF1st23406*	15	Chr2:36758731-36768572	9842	753	250	9.14	28.55
*CpMADS11*	*CpaF1st23696*	9	Chr2:39302712-39326595	23,884	642	213	6.03	25.01
*CpMADS12*	*CpaF1st24126*	8	Chr2:42674312-42685121	10,810	669	222	9.39	25.48
*CpMADS13*	*CpaF1st24137*	5	Chr2:42770150-42773456	3307	537	178	6.12	20.11
*CpMADS14*	*CpaF1st24796*	7	Chr3:5479964-5482602	2639	635	210	8.72	24.62
*CpMADS15*	*CpaF1st25075*	19	Chr3:8607733-8610986	3254	1041	346	6.21	38.64
*CpMADS16*	*CpaF1st28159*	7	Chr4:6573859-6580422	6564	612	203	9.49	23.44
*CpMADS17*	*CpaF1st29740*	8	Chr4:28769871-28783403	13,533	747	248	8.73	28.66
*CpMADS18*	*CpaF1st30512*	18	Chr4:37577074-37585817	8744	1350	449	9.31	50.77
*CpMADS19*	*CpaF1st31440*	8	Chr5:7130147-7137608	7462	687	228	5.91	25.63
*CpMADS20*	*CpaF1st31579*	8	Chr5:8498499-8507170	8672	765	254	8.94	29.36
*CpMADS21*	*CpaF1st33283*	10	Chr5:30392209-30399648	7440	687	228	6.13	25.69
*CpMADS22*	*CpaF1st34609*	8	Chr6:4893230-4907152	13,923	765	254	9.96	28.86
*CpMADS23*	*CpaF1st34610*	16	Chr6:4919734-4928612	8879	738	245	9.04	28.03
*CpMADS24*	*CpaF1st34822*	11	Chr6:6894591-6915492	20,902	780	259	6.06	29.67
*CpMADS25*	*CpaF1st34828*	9	Chr6:6971667-6983725	12,059	474	157	7.73	18.12
*CpMADS26*	*CpaF1st38580*	9	Chr7:21011822-21022154	10,333	732	243	9.43	27.89
*CpMADS27*	*CpaF1st39393*	4	Chr7:29269125-29273783	4659	732	243	7.11	28.22
*CpMADS28*	*CpaF1st39826*	8	Chr7:33119145-33121865	2721	741	246	6.99	28.08
*CpMADS29*	*CpaF1st43513*	10	Chr9:5780448-5787523	7076	891	296	8.51	33.17
*CpMADS30*	*CpaF1st44872*	7	Chr9:24368385-24370857	2473	678	225	9.57	26.00
*CpMADS31*	*CpaF1st04343*	10	Chr10:3501933-3522124	20,192	1254	417	8.89	48.51
*CpMADS32*	*CpaF1st08032*	8	Chr11:12616678-12625690	9013	690	229	9.50	26.22
*CpMADS33*	*CpaF1st11746*	8	Chr12:29355993-29360302	4310	768	255	6.04	29.06
*CpMADS34*	*CpaF1st12042*	8	Chr13:1686053-1691438	5386	870	289	8.85	33.42
*CpMADS35*	*CpaF1st12071*	8	Chr13:1904102-1910372	6271	870	289	7.11	33.15
*CpMADS36*	*CpaF1st12676*	15	Chr13:6960414-6973803	13,390	768	255	6.54	28.89
*CpMADS37*	*CpaF1st12808*	9	Chr13:8140813-8160527	19,715	738	245	7.15	28.29
*CpMADS38*	*CpaF1st12920*	26	Chr13:9350856-9357523	6668	1215	404	6.63	45.74
*CpMADS39*	*CpaF1st13323*	7	Chr13:14729791-14740772	10,982	1134	377	6.73	43.61
*CpMADS40*	*CpaF1st14025*	8	Chr13:25286688-25289400	2713	1116	371	5.23	40.88
*CpMADS41*	*CpaF1st16532*	8	Chr14:26456861-26464798	7938	768	255	6.13	29.05
*CpMADS42*	*CpaF1st18733*	8	Chr16:3824143-3832006	7864	744	247	9.22	28.86
*CpMADS43*	*CpaF1st19998*	14	Chr16:19997527-20003350	5824	687	228	9.22	26.35
*CpMADS44*	*CpaF1st46308*	15	tig00001919:5147437-5153312	5876	783	260	8.71	29.64
*CpMADS45*	*CpaF1st46310*	7	tig00001919:5164160-5180657	16,498	654	217	9.36	25.20

**Table 2 ijms-22-10128-t002:** Analysis of tandem duplication events of MADS-box gene pairs in *C. paliurus*.

Tandem Duplicated Genes	Ka	Ks	Ka/Ks	Purifying Selection
CpaF1st02480 and CpaF1st02481	0.5396	2.4927	0.2165	Yes
CpaF1st23405 and CpaF1st23406	0.5344	2.2205	0.2407	Yes
CpaF1st34609 and CpaF1st34610	1.0263	0.9255	1.1090	No
CpaF1st34822 and CpaF1st34828	1.0524	0.7927	1.3276	No
CpaF1st46308 and CpaF1st46310	0.6368	2.1786	0.2923	Yes

## Data Availability

A total of 18 transcriptome raw datasets (six tissues for triplicates) produced by Illumina NovaSeq 6000 were deposited in the Genome Sequence Archive (SRA) database (https://ngdc.cncb.ac.cn/gsa/s/1b8BK5DP (accessed on 29 August 2021)) under the accession number CRA002980.

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
