# Peer review of "Genome-Wide Identification MIKC-Type MADS-Box Gene Family and Their Roles during Development of Floral Buds in Wheel Wingnut (*Cyclocarya paliurus*)"

_ijms, 2021, doi:10.3390/ijms221810128_

Round 1

Reviewer 1 Report

In this study, Yinquan Qu et al. reported the comprehensive identification and characterization of gene members of MADS-box transcription factor in Cyclocarya paliurus. In total, 45 MIKC-type MADS-box genes were identified and classified into 14 sub-families. Furthermore, systemic analysis of DNA binding motif showed that these MADS-box genes are potentially regulated by multiple transcription factors, indicating their diverse roles. Interestingly, it was showed that most of the newly identified MADS-box genes are relatively enriched in the flower organs of Cyclocarya paliurus, supporting the canonical roles of this family members in regulating the development of flower bud. In summary, this study provides novel genetic basis and insights into the breeding of Cyclocarya paliurus. The experiments are carefully designed and performed to a high standard. Data analysis and results are well presented in a logic manner. The manuscript was well organized and written. I only have some minor concerns:

  1. This study is built on the gene identification by sequence comparison. In the method session, authors should present all the parameters (eg. Command line) used in the program, including BlastP, HMM, HMMER, in detail, and all the criteria used to filtering or selection. The goal of this is to make sure that all the results can be faithfully repeated by other researchers.
  2. The font size should be consistent across all figures (including supplemental figures). For example, gene names and y-axis label in figure1 are too small to be clearly read. Authors should carefully examine and correct all figures and make sure that they meet the requirement of IJMS.
  3. Authors mentioned that they performed RNA-seq to profile the expression levels of MADS-box genes. However, how gene expression is calculated from RNA-seq reads can not be found in the method session. After the reads quality control and adapter trimming, programs and software that are used to mapping reads to the genome and calculate gene expression should be demonstrated.

Reviewer 2 Report

The manuscript describes the identification and expression level of MIKC-type MADS-box  genes, an extensively recurring gene family involved in the floral biology of flowering plants, in Cyclocarya paliurus (Batal.) Iljinskaja. The manuscript is very well written and provides a series of interesting results regarding including the identification in C. paliurus of 45 MIKC-type MADS-box genes and their chromosome location, the length of CpMADS mRNA products  and of the sequences of the consequently translated proteins, analysis of conservation of homologous MIKC-type MADS-box genes between the genomes of C. paliurus and Arabidopsis thaliana, Populus trichocarpa, and Juglans regia and eventually, the expression of MIKC-type MADS-box genes in six different tissues during floral germination via RNA-seq and qRT-PCR.

The analyses were carried out rigorously and the results, although preliminary, provide a solid basis for subsequent studies; therefore I have to congratulate with the authors.
